# Potential Distribution of Wild Host Plants of the Boll Weevil (*Anthonomus* *grandis*) in the United States and Mexico

**DOI:** 10.3390/insects13040337

**Published:** 2022-03-30

**Authors:** Uriel Jeshua Sánchez-Reyes, Robert W. Jones, Tyler J. Raszick, Raul Ruiz-Arce, Gregory A. Sword

**Affiliations:** 1Tecnológico Nacional de México-Instituto Tecnológico de Ciudad Victoria, Boulevard Emilio Portes Gil No. 1301, Ciudad Victoria 87010, Mexico; uriel_elf3@hotmail.com; 2Facultad de Ciencias Naturales, Universidad Autónoma de Querétaro, Avenida de las Ciencias s/n, Juriquilla 76230, Mexico; 3Department of Entomology, Texas A&M University, College Station, TX 77843, USA; tjraszick@tamu.edu (T.J.R.); gasword@tamu.edu (G.A.S.); 4Science & Technology, USDA-APHIS, Edinburg, TX 78541, USA; raul.a.ruiz@usda.gov

**Keywords:** Curculionidae, wild host plants, ecological niche modelling, Gossypieae, conservation

## Abstract

**Simple Summary:**

The boll weevil (*Anthonomus* *grandis* Boheman) (BW) is one of the most well-known and historically important insect pests in the Americas. Presently it is a key pest of cotton in Mexico and Central and South America. In the United States, a successful area-wide pest management program has eradicated the pest from nearly the entire country, except for the Lower Rio Grande Valley region of extreme southern Texas. Eradication in the US has been successful, in part, because the insect can only develop on a narrow range of host plant species besides cotton, none of which occur outside of Texas and Arizona. However, wild host plants are an important consideration for management of this pest in regions where they are present and for preventing reinfestations in areas where eradication has been successful. Here, we present the first detailed analyses of the potential distributions of all known significant wild host plants of the boll weevil in the United States and Mexico. These analyses will enable management to better evaluate the role of wild host plants in their management areas, improve existing boll weevil eradication strategies, and provide insights into the evolutionary history of this important pest insect.

**Abstract:**

The boll weevil (*Anthonomus* *grandis* Boheman) reproduces on a reported 13 species of wild host plants in North America, two in the United States and 12 in Mexico. The distributions of these plants are of economic importance to pest management and provide insight into the evolutionary history and origin of the BW. However, detailed information regarding the distributions of many of these species is lacking. In this article, we present distribution models for all of the reported significant BW host plants from Mexico and the United States using spatial distribution modelling software. Host plant distributions were divided into two groups: “eastern” and “western.” In Mexico, *Hampea* *nutricia* along the Gulf Coast was the most important of the eastern group, and the wild cottons, *Gossypium* *aridum* and *Gossypium thurberi* were most important in the western group. Other species of *Hampea*, *Gossypium*, and *Cienfuegosia* *rosei* have relatively restricted distributions and are of apparent minimal economic importance. *Cienfuegosia* *drummondii* is the only truly wild host in the southern United States, east of New Mexico. Factors determining potential distributions were variable and indicated that species were present in five vegetation types. Ecological and economic considerations of host plant distributions are discussed, as well as threats to host plant conservation.

## 1. Introduction

Wild relatives of domesticated cultivars are valuable biotic resources for the development of management programs and basic research of pest insects [1,2]. Of concern to management is that they may maintain populations of the pest species, which can be a source of new or recurring infestations that damage cultivated crops. More generally, wild host plants are also key sources for natural enemies, genetic reservoirs of genes for plant resistance to insects and other favorable traits, and can provide critical insight into the factors that shaped the evolution of the cultivated plant and its associated insects [3,4,5,6,7,8]. Both the detrimental and beneficial aspects of wild host plants are exemplified in the case of the diversity and distribution of the wild host plants of the cotton boll weevil (*Anthonomus grandis* Boheman) (BW, Figure 1a).

The last comprehensive review of wild host plants of the cotton boll weevil was by Burke et al. in 1986 [4], and host plant distribution maps presented in the paper were roughly estimated based on collection records. At the time of that paper’s publication, BW populations extended to virtually all cotton growing regions of the United States from California to the southeast. Concurrently, cotton was becoming more extensively cultivated in Mexico and Central America, and the insect had already colonized much of the Brazilian cotton growing areas, though it had yet to invade Paraguay and Argentina [9,10].

Subsequently, an eradication program in the United States has eliminated the pest from nearly all cotton growing areas of the US, except for a small region in extreme southern Texas [11]. By 2020, a 10-county region of the Texas Rio Grande Valley was the only area classified as a boll weevil quarantine zone, within which weevils were still being captured in pheromone traps [11]. The continued success and possible completion of the eradication program in the US has meant a shift in focus from continued eradication activities to the prevention of reinfestations of “clean” areas from infested regions in southern Texas and northern Mexico. A reassessment of wild host plant distributions in North America is needed to better understand the sources of indigenous boll weevil populations in Mexico and the United States and to aid in understanding the population genetic data used to identify source populations, their distributions, and host relationships [12].

A reassessment of wild boll weevil host plants is also needed to evaluate the conservation status of these plant species and the potential benefits they may provide, especially for cotton cultivation in Mexico and Central and South America where the boll weevil remains a key pest in almost all areas where cotton is grown [9,10,13]. In many of these regions, present eradication methodologies are not feasible or are complicated by the cultivation of perennial cotton that maintains weevil populations outside eradication zones [9]. Continued research into these wild host plants and their associated organisms can provide insight into their potential use in boll weevil management, for example as genes for plant resistance or sources of natural enemies. However, several species of wild hosts of the boll weevil in Mexico have been categorized as endangered [14,15], primarily because of urbanization and deforestation in Mexico, which lost 7.8% of its tree cover between 2002 and 2020 [16,17].

The objective of the present work was to evaluate the available data on the distribution of truly wild host plant species of the boll weevil in the United States and Mexico, and generate more precise spatial models to evaluate their potential distribution. The host plants evaluated belong to three genera of the cotton tribe, Gossypieae: *Gossypium* L., *Hampea* Schlechtendal, and *Cienfuegosia* Cavanilles [4,18]. Species within these genera have truly wild populations that have been reported to maintain populations of the boll weevil in the wild and in the absence of cultivated cotton [4,18,19,20,21,22,23]. Only two species are reported from the US: *Cienfuegosia drummondii* (A. Gray) Lewton [24] from the southern gulf coast of Texas and *Gossypium thurberi* Todaro in southern Arizona [4,18,25]. Twelve wild hosts occur in Mexico: *Cienfuegosia rosei* Fryxell, *Gossypium aridum* (Rose & Standley ex Rose) Skovsted, *Gossypium davidsonii* Kellogg, *Gossypium harknessii* Brandegee, *Gossypium laxum* L. Phillips, *Gossypium lobatum* H. Gentry, *Gossypium thurberi* Todara, *Gossypium turneri* Fryxell, *Gossyipium hirsutum* L., *Hampea latifolia* Standley, *Hampea nutricia* Fryxell, *Hampea rovirosae* Standley [4]. Determining the potential distribution of these species will aid in (i) evaluating possible sources and patterns of reinfestation of eradicated zones, (ii) elucidating population genetics and evolutionary patterns of boll weevil divergence, and evolutionary patterns, (iii) provide more precise distributions of wild hosts as possible sources of natural enemies and genes favorable to cotton cultivation, (iv) provide data on the conservation status of these plant species.

## 2. Materials and Methods

### 2.1. Selection of Plant Species Considered as Valid Wild Host Plants of Boll Weevil

Species considered in this work are truly wild host plants found in natural habitats that have been confirmed to support immature development of the boll weevil within buds or fruit. Additionally, this reproduction is not restricted to plants found only in the proximity of cultivated cotton [4,18,19,20,21,22,23]. Presently, these species are all members of the cotton tribe Gossypieae. There are reports of other plant genera serving as host plants (*Thespesia, Hibiscus*, *Pseudoabutilon*, *Sphaeralcea*) [4,18]. However, for species of these genera, although adult BW may have been trapped in the field near other Malvaceae and/or reproduction is reported under laboratory or plant nursery conditions [18], confirmation as reproductive hosts under field conditions has yet to be determined [4]. The situation of *Talipariti pernambucense* (Arruda) Bovini is unique in that reproduction is reported in the flower buds of this plant, but only in the proximity of cultivated cotton and only at a single site in southern Chiapas. It is thus considered a marginal host plant [4,28]. Cultivated varieties or races of *Gossypium hirsutum* are not included in the present work [29] nor is the cultivated species, *Gossypium barbadense* L. [4,30].

### 2.2. Geographical Records

Geographical records were obtained from the Global Biodiversity Information Facility site online [31], as well as from available literature. In addition, field data obtained during the last 30 years were included. Available latitudinal and longitudunal coordinates were validated and revised (when necessary) for all species and obtained for locality-only data whenever possible using Geolocate or Google Earth. Only specimens with valid coordinates were used to inform the models. Dubious records or likely erroneous locations (i.e., located in the sea) were removed prior to modelling.

### 2.3. Environmental Variables

Data were obtained for a total of 23 bioclimatic and elevational variables. Nineteen were obtained from WorldClim version 2.1 online site, which considered average conditions for the years 1970–2000 [32]; in addition, elevation, annual evapotranspiration, aridity index [33], and soil type [34] data were also included. Each variable consisted of a raster file with a 30 arcsecond resolution (each pixel representing 1 km^2^). Geographic coordinates of all species were used to extract the data for all 23 environmental layers, using ArcMap 10.2 [35]. Extracted environmental data were assessed with a principal component analysis (PCA) in STATISTICA 8.0 to detect those variables that were both highly correlated and with a low contribution to the multivariate ordination [15,36]. Based on this analysis, only 10 variables were considered appropriate for modeling potential distributions: mean diurnal range (°C), isothermality (%), minimum temperature of coldest month (°C), temperature annual range (°C), precipitation seasonality (coefficient of variation in %), precipitation of wettest quarter (mm), precipitation of driest quarter (mm) (all defined according to O’Donnell and Ignizio [37]), elevation (masl), soil type, and aridity index. Aridity quantifies precipitation availability over atmospheric water demand, and is expressed as a generalized function of precipitation, temperature, and potential evapotranspiration [33]; the index increases for moister conditions, decreases with more arid conditions, and can be best understood using the following scale of five categories [38]: hyper arid (<0.03), arid (0.03–0.2), semi-arid (0.2–0.5), dry sub-humid (0.5–0.65), and humid (>0.65). The selected environmental layers were homogenized prior to niche modelling to achieve the same number of rows, columns, and geographic limits between layers; processing of environmental layers was conducted in ArcMap 10.2.

### 2.4. Niche Modelling

Potential distribution was modelled using the Maximum Entropy Species Distribution Modelling Version 3.3.3k (MAXENT) software. Ten replicates were generated for each species by selecting a random seed testing value of 20%, with a replicated run of the subsample type and 1000 maximum iterations [15,39]. Logistic format, response curves, and jackknife analysis were chosen as output options, while extrapolate projection and clamping options were deactivated [40,41]. All remaining settings were used as default in the software [15].

As a result, an average map of the 10 models with the probability of suitable habitat was generated for each species, with total pixel values equaling the unity value [40]. Therefore, the average map file was transformed into a binary file, with only presence (1) and absence (0) values, in IDRISI 17.0 software. As a threshold limit for delimiting presence of species, the average of the 10th percentile value obtained for the 10 individual maps was selected [15,42]. Additionally, the value of area under the curve (AUC), which indicates the precision of the model, was averaged from the 10 individual models for each species [41]; values higher than 0.9 were considered as excellent [40]. Contribution of the variables to the distribution of the species was obtained from the Jackknife analysis in MAXENT, and corresponding plots of the three most important variables for each species were interpreted using the average of the 10th percentile value threshold previously calculated.

## 3. Results

### 3.1. Cienfuegosia drummondii (A. Gray) Lewton

Geographic records and potential distribution of *C*. *drummondii* were restricted only to the southeastern coast of Texas, and it was not predicted to occur in Mexico (Figure 2a). Elevation was the main variable for modeling the potential distribution of this species, whose presence was associated with areas from 0 to 100 masl. The soil type categories Hemic Histosols and Calcic Histosols were the second most important variables. Additionally, minimum temperature of the coldest month in areas with potential distribution of *C*. *drummondii* ranged from approximately 5.5 to 11.5 °C.

### 3.2. Cienfuegosia rosei Fryxell

This species had the smallest distribution of all wild hosts. It is endemic to Mexico, occupying only 0.08% of the Mexican territory and restricted to the low stature, tropical deciduous forests of the wind-swept lowlands of southeastern Oaxaca (Figure 2b). Its potential distribution was related with a very limited range of minimum temperature of coldest month and of annual mean temperature, both variables from 16 to 18 °C. The third variable of importance for this species, precipitation seasonality, ranged from 110 to 130% indicating a relatively arid habitat with rainfall restricted to only a few months of the year.

### 3.3. Gossypium aridum (Rose & Standl.) Skovst

This species had the greatest latitudinal distribution of all species found, from 26° 45′ in northern Sinaloa to 16°30′ in the southern Isthmus of Tehuantepec, a distance of approximately 2000 km (Figure 2c). It also had the second greatest extension of area with a potential distribution of 3.68% of the total area of Mexico. The distribution of the species is mainly within the western states of the country, forming a patchy distribution along the western coast, extending to Oaxaca in the south, with some inland extensions in Colima, Jalisco, Michoacán, Guerrero, Puebla, and Morelos (Figure 2c). A few geographical records and very limited estimates of potential distribution were observed in central Veracruz, on the eastern coast of Mexico. The most important variable for this species was the precipitation seasonality, varying from 105 to 125%, followed by a precipitation of the driest quarter from 0 to 15 mm, and a minimum temperature of coldest month from 9 to 18 °C.

### 3.4. Gossypium davidsonii Kellogg

The potential distribution of *G*. *davidsonii* was 1.03% of the Mexican territory, restricted to the southernmost region of Baja California Sur and extending to the north along the eastern coast of the state (Figure 2d). The most important variable for its potential distribution was the precipitation of the driest quarter, with values ranging from 0 to less than 10 mm. The second most important variable was a precipitation seasonality of 120 to 175%. Following these, the minimum temperature for the coldest month oscillated from approximately 7.5 to 12 °C.

### 3.5. Gossypium harknessii Brandegee

*G*. *harknessii* was also restricted to Baja California Sur, but had a smaller distribution when compared to *G. davidsonii*. *G. harknessii* occupied only 0.58% of the Mexican territory, mainly in the southwestern and central regions of the state, but also in some small fragments along the eastern coast (Figure 2e). Like *G. davidsonii*, the most important variable for its potential distribution was the precipitation of the driest quarter, with values ranging from 0 to less than 10 mm. The second most important variable was the aridity index, ranging from 0 to 0.2. Following these, minimum temperature for the coldest month oscillated from approximately 7.5 to 12 °C.

### 3.6. Gossypium laxum L. Phillips

Geographical records and the principal areas of potential distribution of this species were observed in the northernmost region of Guerrero state, although small fragments of possible suitable habitat also extended into parts of Michoacán, Morelos, Puebla, and Oaxaca, totaling an occupied area of 0.54% of the Mexican territory (Figure 2f). The most important variable for its distribution was precipitation of the driest quarter, with values ranging from 5 to 20 mm, followed by an isothermality from 65 to 71% and two soil type categories, Vertic Cambisols and Solodic Planosols.

### 3.7. Gossypium hirsutum L.

Truly wild populations of this widely cultivated species have been delimited using genetic analysis [29]. These populations are restricted to small areas of the Mexican Gulf Coast, northern Yucatán, and southern Florida, where only a few records have been registered. Potential distribution indicated a very limited presence only in the southernmost areas of Florida and the Keys in the USA (although not a recorded BW host there), the central eastern coastal and southeastern areas of Tamaulipas, northern areas of Veracruz (Figure 3a), and northern areas of the Yucatan Peninsula (Figure 3b). The main variable restricting potential distribution was elevation, with the species being associated with lowland coastal areas lower than 50 masl. Other important variables for wild *G*. *hirsutum* were the minimum temperature of the coldest month, with the species occurring in areas with values of 11 °C and 16 to 17.5 °C, and an annual mean diurnal range varying from 5 to 11.5 °C.

Predicting natural distribution of *G. hirsutum* based on herbarium records is of little value for the present study because feral and cultivated *G*. *hirsutum* are found throughout Mexico and southern Texas (though efforts are being made to eliminate feral populations in Texas), at least at lower elevations [4,29,30]. Plants are often grown as ornamentals and are frequent along the roadside in some regions. These can be older Pre-Columbian domesticated lineages or newer commercial varieties [29].

### 3.8. Gossypium lobatum H. Gentry

This species is closely related geographically and bioclimatically to *G*. *laxum*, with its principle distribution in Michoacán but extended in some patches to Guerrero and Colima, encompassing 0.88% of the Mexican territory (Figure 4a). Precipitation of the driest quarter from 5 to 20 mm, as well as soil types Vertic Cambisols, Leptic Luvisols, and Haplic Planosols, were the most important variables for potential distribution of this species. Annual temperature range was also determined to be a variable of importance for the presence of *G*. *lobatum*, ranging from 22 to 25 °C.

### 3.9. Gossypium thurberi Todara

This species had the third largest area of potential distribution in Mexico, behind that of *H*. *nutricia* and *G*. *aridum*, and is the only cotton species naturally occurring in the United States, found in southern Arizona below 34° latitude. This species has been reported in adjacent regions in the northern and northeastern regions of Sonora, as well as in small patches of western Chihuahua. The total area occupied was 3.27% of the Mexican territory. The potential distribution of the species extended to the southwestern states of the US into Arizona, California, and New Mexico (Figure 4b), although it was not reported from these latter two states. This potential distribution was constrained by precipitation of the driest quarter as observed in other species of this genus in Mexico, although the range in precipitation for *G*. *thurberi* was greatest, varying from 20 to 50 mm. The other two variables of importance were related to variation in temperature, principally the annual temperature range, which fluctuated from 31 to 39.5 °C, and the isothermality from 43 to 57.5%.

### 3.10. Gossypium turneri Fryxell

This relatively recently described species [43] had the second smallest potential distribution in Mexico of the host plants in this study, occupying only the 0.31% of the total country area. It is found along the western coast of the state of Sonora north of Guaymas at 110°57′ to 111°54′ latitude (Figure 4c). Its distribution was related principally with precipitation of the driest quarter of 0 to 10 mm, an aridity index of 0.1 to 0.125, and a precipitation seasonality of 111 to 171%. Although the potential distribution suggested some extension inland, all present collections have been made close to the shoreline.

### 3.11. Hampea latifolia Standley

This is a relatively rare species that only occupied 0.66% of the Mexican territory and was exclusively observed on the southern coast of Chiapas, with a small area of potential distribution extending into Guatemala (Figure 4d). Precipitation of the wettest quarter was the most important variable for potential distribution of *H*. *latifolia*, which was present in areas with values of tropical, wet conditions from 1120 to 1900 mm. The other two principal bioclimatic predictors were the minimum temperature of the coldest month from 12.5 to 21 °C, and the annual temperature range, which varied from 12 to 21 °C.

### 3.12. Hampea nutricia Fryxell

This was the host species with the largest potential distribution of all host plants, encompassing a total of 3.81% of the Mexican territory, extending from 17° to almost 22° north latitude along the lowlands of the Gulf coast. This distribution included the states of Veracruz and Tabasco as well as northern areas of Oaxaca and Chiapas and eastern Puebla (Figure 4e). The presence of the species was principally associated with the variation in the precipitation of the wettest quarter, with values from 700 to 1800 mm.

### 3.13. Hampea rovirosae Standley

This species had a sympatric distribution with *H*. *nutricia*. However, its distribution was considerably less, covering 1.95% of the total country area (Figure 4f). The potential distribution was a continuous area in the southernmost coastal areas of Veracruz and almost all of the state of Tabasco, but it did not extend beyond 95° latitude north in Veracruz. Other small fragments were observed in northern Oaxaca and on the southern coast of Chiapas, although collections were reported only for the latter. The distribution was related mainly with the minimum temperature of coldest month of 15.5 to 18 °C, a precipitation of the wettest quarter of 650 to 2000 mm, and an annual temperature range varying from 14 to 20 °C.

## 4. Discussion

Wild hosts plants of the boll weevil in Mexico and the United States can be divided into two separate groups based on general geographic distributions: the “eastern” group and the “western” group. The eastern group includes all species south of 17° latitude, including the Isthmus of Tehuantepec of Oaxaca, Chiapas, Tabasco, and the peninsula of Yucatan, north along the Gulf Coast of Veracruz to the southern Gulf Coast of Texas. This group has its greatest diversity of hosts and area of occurrence in southern Mexico, which include: *H*. *nutricia*, *H*. *rovirosae*, *H*. *latifolia*, *C*. *rosei*, *C. drummondii, G*. *aridum* (Tehuantepec population), and *G*. *hirsutum*. Of these, the potential distributions of *H*. *rovirosae*, *H*. *latifolia*, and *C*. *rosei* are restricted to southern Mexico, whereas *H. nutricia* and wild *G. hirsutum* are found in southern Mexico as well as north of 19° latitude along the Gulf coast (Figure 2b, Figure 3a and Figure 4d–f). *Cienfuegosia drummondii* has the most northern distribution of this group and is found exclusively along the Gulf coast in southern Texas (Figure 2a). The western group is restricted to the Pacific Coast north of 17° and west of the Sierra Madre Occidental, from Guerrero to southern Arizona, including Baja California. This group is represented exclusively by various species of diploid *Gossypium* (D species) of the cotton tribe subgenus *Houzingenia* [14] and include *G. aridum* (northern population)*, G. laxum, G. lobatum*, *G. thurberi, G. harknessii, G. davidsonii*, and *G. turneri*. Other than feral or cultivated *G. hirsutum,* no wild host plants have been recorded nor did potential distributions extend into the central plateau of Mexico, north of 20° N.

These two groups of host plants are also associated with two correspondingly distinct and diverse genetic lineages of the BW [12,22,23]. These lineages have variously been considered “subspecies” [23,44,45], “forms” [4], “groups” [22], or “lineages” [12]. All weevil populations utilizing host plants of the “eastern group” belong to the distinct but diverse genetic lineage of populations that is the “boll weevil” [46] or the “subspecies *Anthonomus grandis grandis*” [47]. Besides utilizing the wild hosts of the eastern group, this lineage is also the pest of cultivated cotton in all of the southern United States, south and northeastern Mexico, and South America [12,22,23]. The other lineage was originally thought to only utilize *G*. *thurberi* as a host plant and was given the subspecies name *Anthonomus*
*grandis thurberiae* Pierce [47] or “thurberia boll weevil.” However, recent genetic studies of populations from various host plants in Mexico and Arizona, including cultivated cotton, indicate that there are insufficient genetic differences within this lineage to merit a designation based solely on host plant use [12,22,23]. Alvarado et al. [23] suggest using the name “western boll weevil” to better describe the lineage previously designated as *A*. *g*. *thurberiae*.

Both the eastern and western lineages of *A. grandis* display high genetic diversity and strong population structure within each lineage. This may be expected of relatively isolated populations occurring in allopatry on diverse alternative host plants. Within the eastern lineage (ostensibly *A. g. grandis*), Alvarado et al. [23] found that seven of the nine studied populations from various host plants of southern Mexico had high diversity (*h* values; [48]). However, other than the clustering of populations on wild cotton in the Yucatan peninsula, no clear pattern of genetic differentiation was evident between host plant populations. Raszick et al. [12], however, identified at least three distinct genetic groups within the eastern lineage and found that genetic distance between populations was consistent with a pattern of isolation by geographic distance. In the western lineage, a similar case occurred; genetic differences were detected between populations utilizing *G. thurberi* and those using cultivated cotton, but these populations were again also geographically distant. Thus, the evolutionary effects of alternative host plant use on genetic divergence may be confounded by geography, except perhaps in cases where alternative hosts exist in sympatry [12,49]. Nevertheless, boll weevil populations maintain high diversity and may experience somewhat frequent population replacement. These signals are indicative of significant dispersal capabilities, though dispersal events may be punctuated in their frequency, regardless of whether the host plants are wild or cultivated. In addition, present distribution patterns of the eastern lineage in Mexico may be due to human-aided mixing of populations as the result of widespread cultivation and trade of cotton from Pre-Columbian to modern times [50,51]. It is important to note that South American populations have very low genetic diversity, indicative of an apparent founding event in the late 20th century [52,53]. Haplotypes in Brazil were found to be closest to those in northeastern Mexico and southern Texas, the site of the original founder event of the expansion into the southeastern US [53].

Of the eastern group of wild host plants, *Hampea nutricia* had the greatest predicted area of distribution, extending along the Gulf coast from northern Chiapas to northern Veracruz (Figure 2e). Considered the possible ancestral host of the boll weevil [4,45], *H. nutricia* may also be the wild host plant that maintains the largest populations of BW. The species grows to be small to medium-sized trees of the subcanopy in tropical, humid forests (Figure 1c), and it is also common in secondary growth and forest margins [19,54]. This is a dioecious species and only male flower buds are infested by BW, whereas female buds and fruits escape damage [19,55,56]. Individual male trees can support large weevil populations [19], a result of high densities of male flower buds per branch (Figure 1d). Bud infestation rates by BW can often surpass 50% at the height of the rainy season in August and September in Tabasco, Mexico [19]. *Hampea rovirosae* has a smaller but sympatric distribution compared to *H. nutricia,* although it has lower flower bud densities [56] and less is known of infestation rates and ecology pertaining to BW. Other wild hosts of this group from southern Mexico (*H*. *latifolia*, *C*. *rosei*, *G*. *aridum*) have small potential distributions and presumably small populations of BW (Figure 2b,c and Figure 4d). Likewise, wild *G. hirsutum* is also restricted in distribution along the northern coast of Yucatan and the Gulf coast of central Tamaulipas (Figure 1b and Figure 3a,b), with correspondingly small populations of BW [57].

Of the western group of host plants, the most important in economic terms is *G*. *thurberi*, which is well documented within its distribution from southern Arizona to approximately 27° N latitude in western Sonora (Figure 4b). Several records are found outside of the predicted distribution (Figure 4b), presumably due to the presence of the species in specific microhabitats along streams in more humid canyons and washes within extensive arid habitats of Sonora (Figure 1f [54,56]). The distribution of *G*. *thurberi* places it within the same latitude and relative proximity to commercial cotton cultivation in Sonora, as well as in Arizona.

Of the diploid cottons, the arid adapted *G*. *aridum* has the greatest latitudinal distributions of all host plants, extending from the Isthmus of Tehuantepec at 15° latitude, to northern Sinaloa at 26° and almost in sympatry with *G*. *thurberi* (Figure 2c and Figure 4b). This species can reach considerable size (Figure 1h). This is the only plant species reported to be a host of both the eastern and western lineages of BW [22,23], maintaining populations of the eastern lineage (“boll weevil”) in the Isthmus of Tehuantepec, as well as populations of the western boll weevil north of Colima at 19° latitude. No genetic analysis has been conducted of BW from *G*. *aridum* growing within the extensive area between Colima and the Isthmus of Tehuantepec, and little is known of the ecology of the association of this host with BW.

Five other cotton species occur within the latitudinal gradient of *G*. *aridum*: *G*. *turneri*, *G*. *laxum*, *G*. *lobatum*, *G*. *davidsonii*, and *G*. *harknessii* [14,43]. *Gossypium turneri* occurs in Sonora but is highly restricted to its coastal habitat, though it is not uncommon within its limited distribution (Figure 1e; [43]). Populations of this species are known to maintain western boll weevil populations [22,23]. The rare cotton species, *G*. *laxum* and *G*. *lobatum*, have not been sampled for weevils in over 40 years, nor have weevils from these hosts been sequenced. The two *Gossypium* species in Baja California del Sur, *G*. *davidsonii* and *G*. *harknessii*, are relatively well collected and sympatric in distribution. These continue to maintain populations of *A*. *grandis* that have genetic similarities to the western boll weevil [22].

Based on their potential distributions, wild host plant species of the boll weevil occupy five general vegetation types [58]: tropical perennial forest, (*H. nutricia*, *H. rovirosae*, *H. latifolia*); dry tropical deciduous forests (*G. aridum*, *G. laxum*, *G. lobatum*, *C. rosei*); thorn forest (*G. thurberi*); arid scrub (*G. harknessii*, *G. davidsonii*); and coastal strand vegetation (*G. hirsutum*, *G. turneri*). All habitats have marked seasonality. This seasonality also defines the timing of the flowering and fruiting cycles of the host plants [19,28,54,56,59], thus limiting periods of host plant location and reproduction by the BW and obligating individuals to survive extended host-free periods [19]. Even for *A. grandis* associated with *H. nutricia* in tropical perennial forests, often considered the least seasonal vegetation, flowering is limited to three summer months, and individuals must survive a nine month host-free period [19]. Where weevils on wild hosts spend host-free periods is unknown, but as for weevils on cultivated cotton, this period of dormancy is a considered a critical part of the insect’s life cycle [60]. Given the wide variation of climatic conditions and phenological patterns of the varied wild host plants, the physiological and behavioral processes involved in entering and leaving dormancy are probably also highly varied, and as evidenced from studies of BW in cultivated systems [61,62,63] are more dependent on host plant cues than abiotic factors. Understanding the cues governing dormancy in BW on wild host plants may give greater insight into those used by BW in cultivated cotton.

The present economic importance of wild host plants in Mexico as a source of boll weevils to cultivated cotton in southern Texas and in northeastern Mexico is apparently minimal. Only *H*. *nutricia* in northern Veracruz and truly wild *G*. *hirsutum* on the coast of Tamaulipas are presently reported from this general region. The northern limit of the potential distribution of *H*. *nutricia* is 21° 55′ N latitude on the Gulf coast of the state of Veracruz, 38 km south of Tampico, Tamaulipas, which is greater than 350 km from the closest regions presently cultivating cotton in northern Mexico, including northern Tamaulipas and the Laguna Comarca region of Coahuila and Durango. As for wild *G*. *hirsutum* along the northeastern coast in the state of Tamaulipas, the current status of these populations is unknown. However, Jones et al. [57] reported only scattered and dispersed distributions of this wild race among the littoral vegetation of the Laguna Madre region (Figure 1b), although some individuals had BW. Given present habitat destruction and recent hurricane activity, it may be that these populations have not increased. However, at this time, security conditions currently limit the ability to evaluate wild cotton and weevil populations in the region.

The wild cotton species *G. thurberi* and *G. turneri* may serve as sources of boll weevil in the cotton producing regions of northwestern Chihuahua and the Yanqui Valley of Sonora [64,65]. The municipalities of Janos and Casas Grandes of Chihuahua are cotton growing regions and are less than 50 km from the potential distribution and records of *G. thurberi*. It is unknown if these host plants harbor boll weevils, but they could be of concern if eradication programs were conducted in this region. This is also the situation in the Yanqui Valley in Sonora, which has limited cotton production (<5000 ha) [64,65] and is less than 100 km from records and the potential distribution of *G. thurberi* to the east and *G. turneri* to the northwest. The only other region where cotton is grown in northern Mexico is the Mexicali Valley of Baja, California [65], which is approximately 400 km from populations of the nearest wild host, *G. thurberi.*

The only confirmed wild host of the boll weevil in the United States east of New Mexico is *C*. *drummondii*. Given the potential importance of a wild host plant in an eradication zone, it is notable that only two published studies of the association of the boll weevil with this plant were found: Burke and Clark [25] and Coleman et al. [66]. Although *C*. *drummondii* is historically considered a “rare” plant [67], Burke and Clark [25] were able to collect a total of 4247 fruit capsules and 900 flower buds from several localities in Refugio, San Patricio, Neuces, and Kleberg counties of Texas over a period of three years. Of these, 13.7% the fruit capsules and 18.3% of the buds were infested with boll weevil larvae. In 1997, Coleman et al. [66] also found natural infestations of the boll weevil on this host in the San Patricio County of the Coastal Bend of Texas and resident weevil populations colonized and reproduced on *C*. *drummondii* when planted in the Rio Grande Valley. The key question is whether *C*. *drummondii* is a population sink [68] and, as such, unable to maintain a population of the boll weevil without sustained immigration from cultivated or feral cotton. If so, then populations of BW on *C*. *drummondii* have possibly been extirpated as a result of the eradication of immigrating BWs that had previously maintained populations on this host plant. The source versus sink status of *C*. *drummondii* as a BW host plant in Texas warrants further study.

Presently, eight species of *Gossypium* that are host plants of populations of *A*. *grandis* are officially listed within a protection category by the Mexican government [69]. These include: *G*. *aridum*, *G*. *davidsonii*, and *G*. *hirsutum* (Species subject to special protection; “Sujeto a protección especial”); *G*. *lobatum*, *G*. *thurberi*, and *G*. *laxum* (Threatened; “Amenazada”); and *G*. *turneri* and *G*. *harknessii* (Endangered of extinction; “Peligro de Extinction”). Another boll weevil host, *Hampea latifolia*, although not officially listed, was categorized as threatened (“Amenazada”) using standardized criterion of SEMARNAT (2010) [69]. These threatened or endangered species also maintain unique populations of *A*. *grandis* that can also be assumed to be threatened or endangered. These populations can provide valuable insight into the population structure and the sources of weevils in other regions, as well as to the evolutionary history of this pest [12,22,23]. Additionally, wild hosts have also been found to maintain a high diversity of natural enemies of the boll weevil, many of which have not been studied [20]. These associations would also be lost if the wild host and its weevil populations were to become extinct and provides further justification for in situ conservation of the plant species of the cotton tribe in Mexico [15].

## 5. Conclusions

The boll weevil, as for the great majority of anthonomine weevils [70], is oligophagous and restricted to oviposition and immature development on a limited number of host plants. This specificity is one of the keys to the success of eradication programs in the southern US, in that it allowed management of boll weevil populations to be exclusively focused on cultivated cotton without having to contend with populations from other sources. The fact that the eradication has been highly successful, and no recent reinfestations have occurred outside of southern Texas, argues strongly that no reproductive wild host plant of the boll weevil exists in the southeastern US and Central Texas.

Mexico, however, has important wild reproductive host plants of the boll weevil, all of which are apparently members of the cotton tribe [4]. In economic terms, all data indicate that there are no host plants of economic importance other than feral or cultivated *G. hirsutum* in northeastern Mexico. The wild cotton species *G. thurberi* and *G. turneri* in Chihuahua are relatively distant from cultivated cotton, and would only be of concern if eradication programs were implemented. Thus, the minimal importance of wild boll weevil host plants in northern Mexico places emphasis on the implementation of well-established pest management and eradication methodologies, together with efforts to locate and manage feral and escaped domesticated cotton in south Texas and northern Mexico [71,72]. These basic measures are requisite to prevent re-infestation of eradicated zones in the United States, as well as in boll weevil integrated pest management programs for regions producing cotton in Mexico and South America.

## Figures and Tables

**Figure 1 insects-13-00337-f001:**
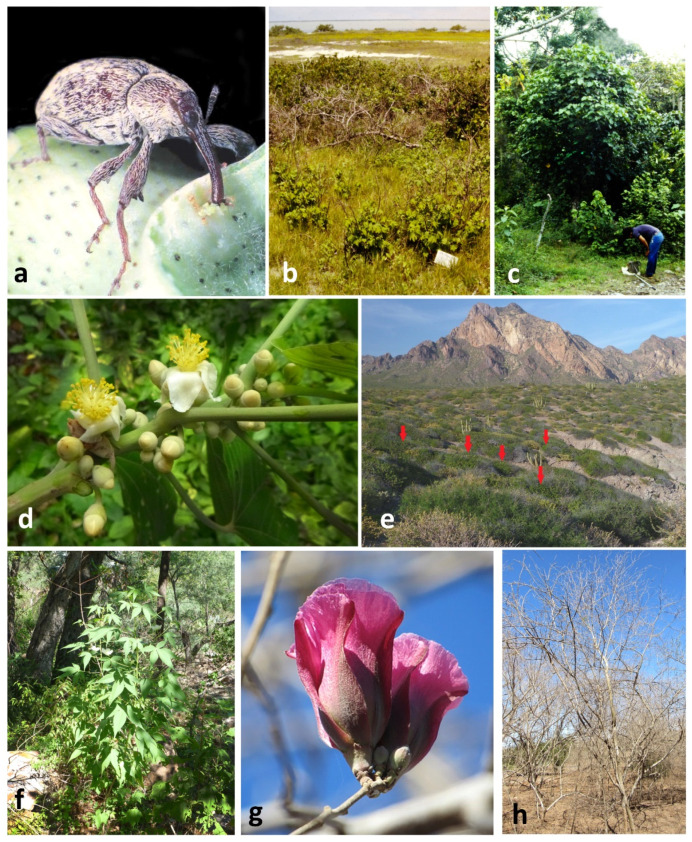
(**a**) The boll weevil, *Anthonomus grandis* Boheman (Photo: Winfield Sterling); (**b**) Wild *Gossypim hirsutum* (foreground) in strand vegetation on shores of the Laguna Madre of Tamaulipas, Mexico (Photo: R. W. Jones); (**c**) Medium sized tree (center) of *Hampea nutricia* Fryxell near Poza Rica, Veracruz, Mexico (Photo: R. W. Jones); (**d**) Branch of *H. nutricia* showing multiple male flower buds per leaf axil; all potential boll weevil oviposition sites (Photo: R. W. Jones); (**e**) Habitat of *Gossypium turneri* Fryxell near shoreline of Pacific Ocean in Sonora, arrows indicating cotton plants (Photo: R. W. Jones); (**f**) *Gossypium thurberi* Todara in isolated canyon northeast of Sahuaripa, Sonora, Mexico (Photo: R. W. Jones); (**g**) Flower buds and flower of *Gossyium aridum* (Rose & Standley ex Rose) Skovsted (Photo: CC © Francisco Miguel Farriols Estrada, By-NC) [26]; (**h**) Deciduous small tree of *G. aridum* during dry season in Sinaloa, Mexico (Photo: CC © Francisco Miguel Farriols Estrada, By-NC) [27].

**Figure 2 insects-13-00337-f002:**
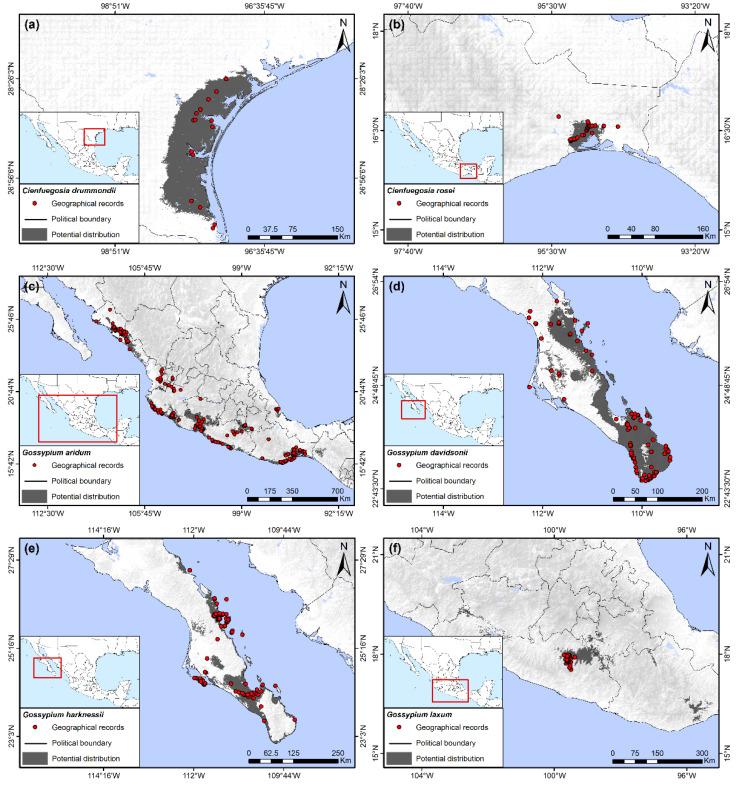
Potential distribution of wild host plants of the boll weevil. (**a**) *Cienfuegosia drummondii*; (**b**) *Cienfuegosia rosei*; (**c**) *Gossypium aridum*; (**d**) *Gossypium davidsonii*; (**e**) *Gossypium harknessii*; (**f**) *Gossypium laxum*.

**Figure 3 insects-13-00337-f003:**
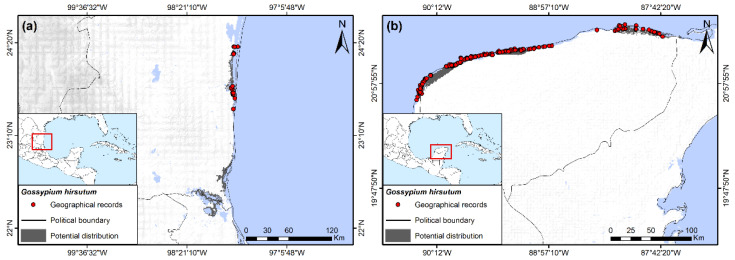
Potential distribution of truly wild *Gossypium hirsutum*, (**a**) Overall distribution along Atlantic coast in South Tamaulipas and northern Veracruz, Mexico; (**b**) Yucatan Peninsula, Mexico (based in part on data from [29]).

**Figure 4 insects-13-00337-f004:**
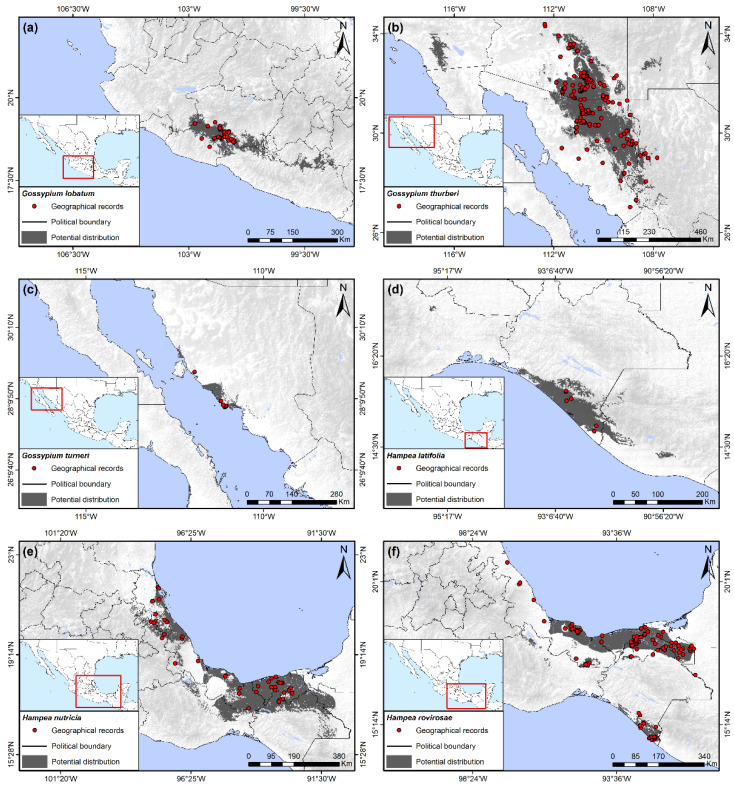
Potential distribution of wild host plants of the boll weevil. (**a**) *Gossypium lobatum*; (**b**) *Gossypium thurberi*; (**c**) *Gossypium turneri*; (**d**) *Hampea latifolia*; (**e**) *Hampea nutricia*; (**f**) *Hampea rovirosae*.

## Data Availability

Data is available through the authors.

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
