# Peer review of "Potential Distribution of Wild Host Plants of the Boll Weevil (Anthonomus grandis) in the United States and Mexico"

_insects, 2022, doi:10.3390/insects13040337_

Round 1
Reviewer 1 Report
The presented article contains a lot of important information regarding the influence of wild plants on the spread of insects considered to be pests. The main part of the work was planned and carried out correctly (except for the editorial stage, which requires improvement - numerous minor errors). In my opinion, however, the work requires substantive supplementation in two places.
The authors should discuss the influence of A. grandis genetic diversity on its distribution in more detail. As this may be a factor more important than the availability of potential resin plants.
The Authors should refer to the influence of climatic changes on the dispersion of A. grandis.
Especially this second paragraph would change the regional character of the work and fit in with world-wide research.

Author Response
insects-1621404 Reponse to Reviewer 1
The reviewer states:
“In my opinion, however, the work requires substantive supplementation in two places.”
“The authors should discuss the influence of A. grandis genetic diversity on its distribution in more detail. As this may be a factor more important than the availability of potential resin plants.”
Response. This is a valid point and we developed the following paragraph which follows paragraph 2 of the discussion beginning at (original) line 355.
Both the eastern and western lineages of A. grandis display high genetic diversity and strong population structure within each lineage. This may be expected of relatively isolated populations occurring in allopatry on diverse alternative host plants. Within the eastern lineage (ostensibly A. g. grandis), Alvarez et al. (2017) found that seven of the nine studied populations from various host plants of southern Mexico had high diversity (h values; Goodall-Copestake et al. 2012). However, other than clustering of populations on wild cotton in the Yucatan peninsula, no clear pattern of genetic differentiation was evident between host plant populations. Raszick et al. (2021), however, identified at least 3 distinct genetic groups within the eastern lineage and found that genetic distance between populations was consistent with a pattern of isolation by geographic distance. In the western lineage, a similar case occurred; genetic differences were detected between populations utilizing G. thurberi and those using cultivated cotton, but these populations were again also geographically distant. Thus, the evolutionary effects of alternative host plant use on genetic divergence use may be confounded by geography, except perhaps in cases where alternative hosts exist in sympatry. Nevertheless, boll weevil populations maintain high diversity and may experience somewhat frequent population replacement. These signals are indicative of significant dispersal capabilities, though dispersal events may be punctuated in their frequency, regardless of whether the host plants are wild or cultivated. In addition, present distribution patterns of the eastern lineage in Mexico may be due to human-aided mixing of populations as the result of widespread cultivation and trade of cotton from Pre-Columbian to modern times (Rodríguez Vallejo, 1976, Carrillo Rojas 2013). It is important to note that South American populations have very low genetic diversity, indicative of an apparent founding event in the late 20th century (Marquesini et. al 2021, Martins et al. 2007). Haplotypes in Brazil were found to be closest to those in northeastern Mexico and southern Texas, the site of the original founder event of the expansion into the southeastern US (Marquesini et. al 2021).
The second suggestion is:
The Authors should refer to the influence of climatic changes on the dispersion of A. grandis.
Especially this second paragraph would change the regional character of the work and fit in with world-wide research.
The authors agree that this is an important possible factor in the distribution of the boll weevil, especially for possible future scenarios of invasion of eradicated zones. In fact, two of the authors (Rasick and Sword) are considering the modeling of future climates to evaluate possible changes in distribution and invasion potential of the boll weevil. However, this is presently only a future research project that would require more work. Considering the present research focused on determining present host plant distributions, it seems beyond the scope of the present paper.
Reviewer 2 Report
This paper by Sanchez-Reyes and coauthors considers the distribution of 14 species of host plants of Anthonomus grandis, a significant pest of cotton. The presence of these plants in cotton-growing regions may affect the strategies used to manage A. grandis. Therefore, it is appropriate that this largely plant-focused contribution is to be published in a journal directed to entomologists.
The authors use species distribution modelling to extrapolate the potential distribution of the focal plant species from point distribution records retrieved from GBIF and from field work conducted by the authors. These location data were fitted with a MAXENT model using several bioclimatic variables as predictors.
I have no major criticisms of the paper. The methods have been used appropriately, and the results shown make sense. The paper is well-written with minimal spelling and grammatical mistakes and it is easy to follow.
One suggestion for improving the paper is the inclusion of a combined map of the entire area considered by the authors, which shows the total area covered by wild host plants, highlighting areas (if any) where multiple host species may be found. Overlaying this with areas where the weevil is presently found and where cotton is commonly grown would also be interesting. A map like this would be helpful for interpreting paragraphs like that in lines 428--438.
I also recommend that the authors move the final paragraph (lines 478--492) to the discussion. Although the conservation status of these plants is of interest, the main purpose of this contribution is to assess the importance of these plants in A. grandis management. I believe that the paragraph on line 466--477 is an accurate summary of the authors' findings, and serves as a fitting closing paragraph to the paper. Also, a reiteration of the argument of lines 249--253 in the conclusion (feral/cultivated G. hirsutum are found throughout Mexico and are likely to be the most important reservoirs of A. grandis for commercial cotton growers) would be worthwhile.
In addition, the following minor spelling mistakes were noted:
Line 134. Change "previous" to "prior"
Line 237. Heading. Change "Gossyipium" to "Gossypium"
All maps have the phrase "Politic boundary" in the legend. If possible, I would recommend these be changed to "Political boundary"
Congratulations to the authors on producing a fine paper.
Author Response
insects-1621404 Response to Reviewer 2
The reviewer states:
“One suggestion for improving the paper is the inclusion of a combined map of the entire area considered by the authors, which shows the total area covered by wild host plants, highlighting areas (if any) where multiple host species may be found. Overlaying this with areas where the weevil is presently found and where cotton is commonly grown would also be interesting. A map like this would be helpful for interpreting paragraphs like that in lines 428--438.
This suggestion was considered and we attempted to make the map as requested with different colors highlighting overlap regions. However it didn’t seem to present a clearer understanding of the overlap of the wild host plants and the connections with regions of cultivated cotton. This was for several reasons. Although the map helped visualize all hosts, there were only two areas of major ovelap. The first in in Baja California where Gossypium davidsonii and Gossypium harknessii ovelap. But presently, these happen to be adjacent maps in the paper (2d and 2e) so the overlap is easy to appreciate in the present maps. The same is for the largest area of overlap between Hampea nutricia and Hampea rovirosae where again the present maps are adjacent and of the same scale, (4e and 4f). The only other areas of overlap were Gossypium aridum and Cienfuegosia rosei in the Isthmus of Tehuantepec but the distribution of the latter species is so small that it was difficult to appreciate on the large scale map,
As for lines 428 to 438 the principal limitation of producing a map of this area for northern Mexico is not the wild hosts but the difficulty of producing a reliable map of regions of cotton production in Chihuhua and the surrounding areas. The cultivation of cotton in Chihuahua is not of large acreages and varies from year to year with some increase in recent years (Inforural 2021. Reporte anual de producción de algodón y derivados. Available online: https://www.gob.mx/agricultura/articulos/el-algodon-de-mexico-fibra-suave-y-cultivo-generoso?idiom=es (accessed on 12 January 2022). There is simplly not enough information to produce reliable maps of the commercial cotton production.
The reviewer also states:
I also recommend that the authors move the final paragraph (lines 478--492) to the discussion. Although the conservation status of these plants is of interest, the main purpose of this contribution is to assess the importance of these plants in A. grandis management. I believe that the paragraph on line 466--477 is an accurate summary of the authors' findings, and serves as a fitting closing paragraph to the paper. Also, a reiteration of the argument of lines 249--253 in the conclusion (feral/cultivated G. hirsutum are found throughout Mexico and are likely to be the most important reservoirs of A. grandis for commercial cotton growers) would be worthwhile.
This was an excellent suggestion and the paragraph was moved to the end of the Discussion as suggested.
The following observations were corrected:
In addition, the following minor spelling mistakes were noted:
Line 134. Change "previous" to "prior"
Line 237. Heading. Change "Gossyipium" to "Gossypium"
All maps have the phrase "Politic boundary" in the legend. If possible, I would recommend these be changed to "Political boundary" – All maps were reconfigured as suggested
Reviewer 3 Report
First of all, I want to thank you for the interesting contribution; I find it well written and simple (in a positive way).
There is not much I can point out, except for a couple of phrases that might be re-written and the references that must be revised.

Author Response
insects-1621404 Response to Reviewer 3
Reviewers comments: There is not much I can point out, except for a couple of phrases that might be re-written and the references that must be revised.
All authors made a thorough revision on the text and considerable revisions were made to the wording and the references were rechecked.
Round 2
Reviewer 1 Report
The manuscript in the presented form meets the criteria for printing.I accept the changes made by the Authors. The article meets my expectations.